# Exploiting TLK1 and Cisplatin Synergy for Synthetic Lethality in Androgen-Insensitive Prostate Cancer

**DOI:** 10.3390/biomedicines11112987

**Published:** 2023-11-07

**Authors:** Siddhant Bhoir, Oluwatobi Ogundepo, Xiuping Yu, Runhua Shi, Arrigo De Benedetti

**Affiliations:** 1Department of Biochemistry and Molecular Biology, LSU Health Shreveport, 1501 Kings Hwy, Shreveport, LA 71103, USA; siddhant.bhoir@lsuhs.edu (S.B.); oluwatobi.ogundepo@lsuhs.edu (O.O.); xiuping.yu@lsuhs.edu (X.Y.); 2Department of Medicine, LSU Health Shreveport, 1501 Kings Hwy, Shreveport, LA 71103, USA

**Keywords:** PCa, synthetic lethality, TLK1 signaling, homologous recombination repair, TLK1 inhibitor J54, CPT-based PCa therapy

## Abstract

Cellular organisms possess intricate DNA damage repair and tolerance pathways to manage various DNA lesions arising from endogenous or exogenous sources. The dysregulation of these pathways is associated with cancer development and progression. Synthetic lethality (SL), a promising cancer therapy concept, involves exploiting the simultaneous functional loss of two genes for selective cell death. PARP inhibitors (PARPis) have demonstrated success in BRCA-deficient tumors. Cisplatin (CPT), a widely used chemotherapy agent, forms DNA adducts and crosslinks, rendering it effective against various cancers, but less so for prostate cancer (PCa) due to resistance and toxicity. Here, we explore the therapeutic potential of TLK1, a kinase upregulated in androgen-insensitive PCa cells, as a target for enhancing CPT-based therapy. TLK1 phosphorylates key homologous recombination repair (HRR) proteins, RAD54L and RAD54B, which are critical for HRR alongside RAD51. The combination of CPT with TLK1 inhibitor J54 exhibits SL in androgen-insensitive PCa cells. The formation of double-strand break intermediates during inter-strand crosslink processing necessitates HRR for effective repair. Therefore, targeting TLK1 with J54 enhances the SL of CPT by impeding HRR, leading to increased sensitivity in PCa cells. These findings suggest a promising approach for improving CPT-based therapies in PCa, particularly in androgen-insensitive cases. By elucidating the role of TLK1 in CPT resistance, this study provides valuable insights into potential therapeutic targets to overcome PCa resistance to CPT chemotherapy. Further investigations into TLK1 inhibition in combination with other DNA-damaging agents may pave the way for more effective and targeted treatments for PCa and other cancers that exhibit resistance to traditional chemotherapy agents.

## 1. Introduction

All cellular organisms, including humans, have multiple highly conserved DNA damage repair and tolerance pathways to contend with different DNA lesions that form spontaneously or upon treatment with DNA-damaging agents, including radio- and chemo-therapeutics [1,2]. For example, bulky and/or helix-distorting lesions are repaired through nucleotide excision repair (NER) [3]. On the other hand, non-bulky and non-helix-distorting base lesions, as well as single-strand breaks (SSBs), are repaired through base excision repair (BER) [4]. Double-strand breaks (DSBs) can be repaired via homologous recombination repair (HRR), which is generally error-free, and nonhomologous end-joining (NHEJ), which generally creates insertion/deletion at the damaged sites [5]. DNA lesions can also be tolerated by the so-called translesion synthesis (TLS), catalyzed by specialized and generally error-prone DNA polymerases. The activation of DNA damage checkpoints (DDCs), which temporarily arrest the cell cycle before DNA lesions are repaired, is also an important mechanism for the cell to contend with DNA damage [6]. By maintaining genome stability, most genes involved in DNA damage repair, tolerance and DDC serve as *bona fide* cancer suppressors.

Synthetic lethality (SL), where the functional loss of two genes together, but not alone, results in cell death, is a conceptually new strategy for cancer therapies [7]. The first successful SL treatment strategy is using PARP inhibitors (PARPis) to treat tumors with the functional loss of BRCA1 or BRAC2, two cancer suppressors involved in the HRR of DSBs. PARPis prevent PARP1 and PARP2 from repairing SSBs, leading to stalled and collapsed DNA replication forks. Subsequently, SSBs are converted to DSBs that HRR-deficient cells cannot repair effectively, leading to cell death. Recently, PARPis have been shown to be effective for killing PCa cells with the functional loss of BRCA1, BRCA2 or other proteins that are directly or indirectly involved in HRR, and moderately extend the survival of mCRPC patients with this functional loss [8,9,10].

Cisplatin (CPT) reacts with DNA, producing single purine adducts and intra- and inter-strand crosslinks (ICLs) [11,12,13,14]. The single purine adducts and intra-strand crosslinks can be repaired through NER or tolerated by translesion synthesis (TLS). The repair of inter-strand crosslinks is more complicated, requiring the participation of Fanconi anemia (FA) proteins and subsets of proteins involved in NER, TLS, HRR, NHEJ and BER [15,16]. Therefore, deficiency in any of the DNA damage repair, tolerance and DDC mechanisms may cause CPT sensitivity, and combined deficiencies of two or more of these mechanisms will synergistically enhance this sensitivity.

CPT has been most frequently used for the adjuvant or neoadjuvant treatment of most solid tumors [17]. Impressively, CPT treatment is highly efficient against testicular germ-cell cancer, leading to a durable complete remission in >80% of patients [18]. While CPT-based therapy has not been the treatment of choice for PCa, partly due to significant renal toxicity (which can be faithfully recapitulated in mice [19]), recent studies have indicated that lower dosing in combination with inhibitors of DNA damage repair could be quite effective [14]. If not repaired, the bulky CPT-DNA adducts that are formed cause a block in DNA replication and/or transcription, resulting in apoptosis [20]. However, as multiple DNA damage repair, tolerance and DDC mechanisms are implicated in CPT resistance (Figure 1) [12,21], the genes that can be targeted to improve CPT-based PCa therapies remain largely unexplored. CPT is highly effective against several forms of cancer, most notably testicular tumors, and it is also commonly used to treat breast, ovarian, bladder, lung, and head and neck cancer. However, prostate cancer (PCa) is resistant to CPT chemotherapy due to poor targeting and the development of resistance. Some strategies for targeting CPT by piggybacking carrier nanoparticles onto PSMA [22], as well as recent clinical studies that indicated that lower dosing in combination with DNA damage repair inhibitors (or for cases with defective DNA repair genes) have shown to be quite effective [14,23,24]. However, as multiple DNA damage repair, tolerance and DDC mechanisms are implicated in CPT resistance [12,21], it is not obvious which genes need to be targeted to improve CPT-based therapies for the majority of PCa cases.

TLK1 is a serine/threonine kinase [25] that is frequently upregulated in PCa cells upon anti-androgen therapies or after DNA damage [26,27]. By phosphorylating its substrates, TLK1 plays a role in resistance to several DNA-damaging agents, including CPT [28,29,30,31]. Our recent work showed that TLK1 specifically catalyzes the phosphorylation of the HRR proteins RAD54L and RAD54B [32], which, at least in part, explains the previously reported role for TLK1 in CPT resistance in cholangiocarcinomas [31], as together, these two paralogs carry out some fundamental functions in HRR along with RAD51 [33]. Since the processing of ICLs typically results in the formation of DSB intermediates that require the HRR pathway to complete lesions’ repair, it is expected that the combination of CPT with inhibitors of HRR will result in synthetic lethality. In this work, we tested the effect of a specific inhibitor of TLK1 in combination with CPT as a therapeutic strategy for androgen-insensitive PCa cells.

## 2. Materials and Methods

An RPMI-1640 medium was purchased from thermofisher. CPT was purchased from Millipore-Sigma (P4394). The J54 was synthesized by our group as described in the STAR Methods of [34]. However, we are aware that it is now sold by Probechem (TLK1 inhibitor J54|TLK1 inhibitor|Probechem Biochemicals).

### 2.1. Cell Viability Assay

Cell viability was evaluated using the MTT assay, which measures the reduction of 3-(4, 5-dimethylthiazol-2-yl)-2, 5-diphenyl tetrazolium bromide (MTT) by mitochondrial enzymes in live cells. We seeded 20,000 PC-3 and C4-2B cells into 100 μL of the medium in 96-well plates and allowed them to adhere for 24 h. Then, we replaced the medium with a fresh medium containing various cisplatin and/or J54 concentrations, and incubated them for an additional 24 h. Finally, we added the MTT reagent to each well and incubated it for 35 min before measuring the absorbance intensity at 490 nm using a spectrophotometer.

### 2.2. Animal Studies

All animals used in this study received humane care based on the recommendations set by the American Veterinary Medical Association, and the Institutional Animal Care and Use Committee of the LSU Health Sciences Center at Shreveport approved all the test protocols. Immune-deficient NOD SCID mice (Charles River, Skokie, IL, USA) were used in this research to host human PCa PC-3 tumors; 0.5 × 10^6^ human PCa PC-3-Luc cells suspended in Matrigel were grafted subcutaneously into the two lower back flanks of the NOD SCID mice. After the tumor sizes of ~150 mm^3^ were established, the tumor-bearing mice were randomized into eight treatment groups. A 2 × 4 factorial design was used in the current study. The mice were treated with a vehicle control (PBS, J54, 0 mg/kg), a TLK1-RAD54 axis inhibitor J54 (5 mg/kg), and different dosages of CPT (0, 1, 3, and 6 mg/kg). ntraperitoneal (IP) injections of J54 (dissolved in 200 sterile saline with 10% Polysorbate-80—PS-80) were given bi-weekly. The dose for J54 was based on our previous work [34]. CPT dissolved in a 0.9% sterile-filtered NaCl solution (saline) was administered individually at different dosages (1, 3, and 6 mg/kg) in a 100 uL volume via IP injection twice a week. The tumor sizes were measured every other day using a caliper. The inhibitor treatment lasted 28 days for about nine bi-weekly drug cycles. The tumor-bearing mice were monitored every other day and euthanized if there was an apparent loss in body weight (≥20%), abdominal palpitation due to the development of prostate tumors or cancer metastasis, poor body condition, or the mice were too sick and unable to reach food and water. The mice were sacrificed at the end of the experiment via CO_2_ asphyxiation, and the tumors were excised for the tissue Western blots.

### 2.3. Western Blots

a.Tissue Western Blot: Western blots were performed in three biological replicates for the tumors excised from the different treatment groups, including the control (PBS), cisplatin (3 mg/kg), J54 (5 mg/kg) and the combination of the PC-3 grafted NOD SCID mice. The frozen tumor tissues were disrupted with the Bioruptor^®^ Plus sonication device (Diagenode; Cat. No. B01020001), and homogenized and lyzed in the ice-cold RIPA lysis buffer system (Santa Cruz Biotechnology, Dallas, TX, USA; Cat. No. SC-24948). The samples were clarified via centrifugation at 13,000 rpm for 20 min in the refrigerated setting. The supernatant was collected, transferred into fresh 1.5 mL microfuge tubes, flash-frozen and stored at −80 °C until further use. The total protein concentration was measured using a Pierce™ BCA protein assay kit (Thermo Scientific, Waltham, MA, USA; Cat. No. 23225) with bovine serum albumin (BSA) as a standard control. An equal loading amount of 15 µg was calculated for each protein sample. The sample supernatant was denatured with 1X Laemmli Buffer for 10 min at 950C and separated using 12% Mini PROTEAN TGX protein gel (BioRad, Hercules, CA, USA; Cat. No. 4568084) at 100 volts for 120 min. The proteins were transferred to the Immun-Blot PVDF membrane (BioRad; Cat. No. 1620177) using a Mini Trans-Blot Cell (BioRad; Cat. No. 1703930) at 100 volts for 150–180 min on ice. The membrane was blocked with 5% non-fat dry milk (Cell Signaling Technology, Danvers, MA, USA; Cat. No. 9999S) in 1X Tris-buffered saline with Tween-20 (TBST) for 1 h at room temperature. Following blocking, the membrane was washed once with 1X TBST and incubated with mouse anti-PCNA (PC10) monoclonal antibodies (Santa Cruz Biotechnology; Cat. No. SC-56; 1:1000 dilution) and mouse anti-PARP-1 (F-2) monoclonal antibodies (Santa Cruz Biotechnology; Cat. No. SC-8007; 1:1000 dilution) or anti-Cl-CAS3 (Asp175) rabbit antibodies (Santa Cruz Biotechnology; Cat. No. SC-9661) in 5% BSA in 1X TBST overnight at 4 °C with gentle rocking. The next day, after washing four times with 1X TBST, the membrane was incubated with horse anti-mouse antibodies (Cell Signaling Technology; Cat. No. 7076S: 1:2000 dilution) labeled with horseradish peroxidase in 5% BSA in 1X TBST for 1–1.5 h at room temperature. After incubation, the membrane was washed four times with 1X TBST, and the reactive bands were detected using a Pierce™ ECL Western Blotting Substrate (Thermo Scientific; Cat. No. 32106) on the ChemiDoc MP Imaging System (BioRad; Cat. No. 12003154).b.Cell Western Blot: The Western blot for the PC-3 cells was performed as described above but with minor modifications. Briefly, 3 × 10^6^ PC-3 cells (control and drug-treated) were collected, washed twice with ice-cold PBS and lyzed with the RIPA lysis buffer system. The lysate was vortexed and centrifuged at 13,000 rpm for 10 min to remove cell debris. The total protein was estimated, and 30 μg of the cell lysate was loaded onto an SDS-PAGE gel. The separated proteins were transferred to the membrane using a wet transfer apparatus. The complete transfer was ensured by checking the membrane for uniform background staining. The membrane was then incubated in a blocking solution (e.g., 5% non-fat milk in TBST) for 1 h at room temperature to block non-specific binding sites, followed by primary antibody (custom-made anti-pRAD54 rabbit polyclonal; Thermo Scientific; Cat. No. AB1991; 1:1000) incubation in a blocking solution overnight at 4 °C. The next day, the membrane was washed 3× with TBST for 10 min each to remove excess primary antibodies. Further, it was incubated with goat anti-rabbit HRP-conjugated secondary antibodies diluted in a blocking solution for 1 h at room temperature. Next, the membrane was washed 3× with TBST for 10 min each to remove excess secondary antibodies, and the bands were detected using the ECL chemiluminescent substrate.

### 2.4. Statistical Analysis

The tumor volume and body weight were measured and used for the final analysis, including 22 June 2023–16 July 2023. The tumor weight was analyzed on 4 August 2023 only when the mice were sacrificed. The survival time was computed between the date of death or the last study date (4 August 2023) and the date of treatment (22 June 2023). The area under the curve (AUC) was calculated using the trapezoid method for each mouse. A two-way analysis of variance (ANOVA) was used to assess the effect of cisplatin and J54 on the tumor volume, body weight and AUC. A one-way ANOVA was used while there was an interaction effect between the cisplatin and J54. The life table method was used to estimate the survival for each group, and the Sidak method was used for adjustment for multiple comparisons for the log-rank test. All statistical analyses were performed using Statistical Software SAS 9.4 for Windows (SAS Institute Inc., Gary, NC, USA). All *p*-values < 0.05 were considered to be statistically significant.

### 2.5. Data Availability

A description of all data and materials can be found in the referenced article. No additional data are withheld from the public. Correspondence and requests for data underlying findings or materials should be addressed to ADB.

## 3. Results

Combining CPT with J54 enhances its growth inhibitory potential on CRPC cells. For our work, we selected two CRPC cell lines that utilize different mechanisms of androgen-insensitive growth (PC3 and C4-2B), which are known to be rather resistant to CPT. In Figure 1, we show the viability results, using MTT, of the two cell lines’ cultures over 48 h with cisplatin, either alone or in combination with the TLK inhibitor, J54. It is noticeable that, up to a concentration of 0.3 μg/mL (already quite high for the therapeutic index), both cell lines were quite insensitive to CPT, and in fact, C4-2B even grew slightly better than the untreated cells (possibly an effect of the induction of DNA repair enzymes by CPT that may help in the complex process of DNA replication). As expected, both cell lines display progressive growth inhibition at higher CPT concentrations. The situation, however, was quite different when the experiment was carried out in combination with 5 μM of J54 (known to produce a full inhibitory effect on TLK1). In this case, both cell lines displayed a much stronger, dose-dependent reduction in viability, and the peculiar growth stimulation observed at low doses of CPT for the C4-2B cells was eliminated. As previously reported, J54 alone (0 CPT) had no growth inhibitory effect on these cells up to 13 µM [34]. Thus, while synergism is usually determined using the method of Chou and Talalay [35], whereby the resulting median effect lines and the x-axis intercept (log IC_50_) and slope (m) (a measure of sigmoidicity) are calculated for each drug and combination using the least squares method, the lack of J54 toxicity at 5 μM made it sufficient only to vary CPT.

CPT treatment induces pRAD54-T700. The fundamental hypothesis of our work is that the repair process of ICL lesions induced by CPT results in the formation of transient DSBs that require repair via HRR. Hence, interfering with the mechanism of HRR by preventing the critical phosphorylation of RAD54-T700, mediated specifically by TLK1, will enhance the toxic effects of CPT.

We have recently generated a highly specific pRAD-T700 antiserum that can specifically monitor the activity of TLK1 during HRR [32]. However, this antiserum has not been tested yet on DNA-damaging agents other than IR. We have now observed that CPT is a potent inducer of RAD54 phosphorylation, while the TLK1 inhibitor J54 partly suppresses this. This was independently confirmed in both CRPC PCa cell lines (Figure 2). Since this was also observed for the recovery period after IR [32], it is tempting to speculate that the phosphorylation of RAD54-T700 is either a general phenomenon that occurs with other DNA-damaging agents, or (more likely) that ICLs being converted to DSBs are partly repaired via canonical HRR requiring pRAD54-T700 modification.

The combination of CPT + J54 results in the dose-dependent regression of tumor xenografts. To establish if adding J54 to CPT has an actual therapeutic effect from a synthetic lethal combination with a genotoxic agent, we investigated the effect on the growth of a subcutaneous PC3-Luc tumor model in NOD-SCID mice. A main reason is that PC3 (AR-) displays some features shared by NEPC cells, including the expression of chromogranin A. Whilst, for AR+ cancers (like C4-2B cells), there are some options, NEPC patients have CPT as one of their few remaining treatments. After inoculating 1M cells on both dorsal flanks, the tumors were allowed to form to a size of ~150 mm^3^, after which the treatment started. We used a fixed dose of 5 mg/kg of J54 and three different dosages of CPT (0, 1, 3 and 6 mg/kg). The progression or regression of the tumors was followed in each mouse with calipers, and the average tumor size over time is plotted in Figure 3 in the bottom left panel. Notably, all the treatments were at least cytostatic compared to the controls, while the combination treatments with 1 mg/kg and 3 mg/kg of CPT after 40 days resulted in the cumulative regression of the tumors. When the experiment was concluded, and the tumors were recovered from the group treated with 3 mg/kg of CPT + J54, the remnants were hardly measurable with calipers or by weight. The 6 mg/kg CPT (±J54) dose was also highly effective at cytoreduction, but the treatment could be continued for only six cycles (bi-weekly) before the toxicity was such that it required euthanasia before the conclusion of the experiment, and even before the visualization with the IVIS.

At the experiment midpoint and a day before sacrificing the mice (Figure 3), the animals were injected with luciferin, and the actual tumor cell mass was visualized with an IVIS-Spectrum/CT machine (Perkin Elmer). While the IVIS was run unassisted with auto-acquisition, which allows for the detection of the full range of cells, from very few to very large numbers, resulting in the production of tumor images from all the animals, the details are much better appreciated in the heat-map scale (or from the individual radiance quantitations). For instance, the most remarkable result is seen in the CPT 3 mg/kg + J54 group. While PC3-Luc cell tumors are also seen in this group, the heat-map scale is drastically different: it ranges from 10,000 to 50,000. In contrast, in the control (for example), the scale ranges from 0.2 to 1.2 × 10^11^, meaning up to six logs of radiance units lower in the combination than in the tumors from the control group. For a more precise comparison between the control group and the 3 mg/kg combination, we measured the average flux of all the tumors in each group. For the control, the average flux was 4.61 × 10^10^, while for the combination, it was 31,604. Another interesting observation was that J54 alone affected tumor growth in some animals, for example, very noticeably in the 3d mouse from the left. Although the growth curve measured with calipers is displayed as the average of all the tumors in a group, we did notice a partial cytostatic effect with J54 alone, which became an actual tumor regression toward the late part of the experiment. Another important observation is that J54 not only had no apparent toxic effect and did not affect the mice’s weight and water/food consumption, but showed a partial improvement in the visual health conditions of the mice treated with the higher 3 mg/kg CPT dose. Rather than losing weight, a possible sign of cachexia from tumor burden and treatment toxicity, the animals in the combination groups showed stable weight, and were more active and frequently foraging.

The strong tumor regression in mice treated with CPT + J54 is hallmarked by apoptosis. To establish more firmly the mechanism of tumor regression (or at least cytostasis) in the mice treated with CPT, J54 or the combination, we focused on the group of mice treated with 3 mg/kg of CPT, where the effects were more evident, but the toxicity was tolerable (the mice did not lose much weight and appeared healthy and active).

Immediately after necropsy, all the tumors were weighed, and one from three random mice in each group (from the 3 mg/kg dose) were prepared for Western blot analysis of PARP/cleaved PARP (for the determination of the intra-tumoral extent of apoptosis) and PCNA (for the determination of the fraction of still-proliferating cells). As illustrated in Figure 4, it is immediately clear that all the mice treated with CPT alone displayed reduced levels of full-length (Fl) PARP and an array of its cleavage products, with an abundant ~90 kDa product (cl-PARP1) that is considered to be its classic apoptotic signature [36], but smaller processed products were also generated, as indicated in Figure 4. In the CPT + J54 combination treatment, and most prominently in the right-most tumor, the Fl-PARP was strongly reduced, and there was a prominent formation of cl-PARP2 and cl-PARP-3, and entirely bypassing cl-PARP1, suggesting a much more advanced stage of apoptosis and loss of PARP integrity (PARP degradation is mediated by a caspase cascade [37]). It should be noted that the J54 treatment alone also resulted in some cleaved PARP products, consistent with our previous report that its related compound (Thioridazine also acting as an inhibitor of TLK1), while not toxic for PC3 cells in vitro, resulted in some tumor regression with distinctive apoptotic signatures when administered in xenografts due to the partly hypoxic tumor microenvironment that tends to generate DNA lesions and the activation of the DDR [38]. In that study, we also extensively characterized the mechanism of enhanced cell death via induced apoptosis when PC3 cells were treated with a combination of doxorubicin (generates DSBs) and Thioridazine that we suggest is analogous to the mechanism seen here for the combination with CPT, although the mechanisms of DNA damage repair for doxorubicin or IR (primarily NHEJ) vs. CPT (primarily NER combined with HRR) are substantially different. While we did not yet test J54 in combination with radiotherapy (IR) we expect some therapy potentiation from the fraction of cells that repair the IR-generated DSBs via HRR.

To confirm the results from the cl-PARP as an indicator of apoptosis, tumors from two additional sets of mice were investigated for the presence of cleaved Caspase 3 (Cl-CAS3; Appendix A), which confirmed that the combination CPT + J54 was required to obtain maximal Cl-CAS3. We should mention that we did not carry out other confirmatory assays of apoptosis, like TUNEL/IHC, because several residual tumors, particularly in the most significant 3 mg/kg CPT + J54 group, were very small, and we had to choose between preparation for WBs or PEFF sections.

We also investigated the status of pRAD54-T700 in these tumor extracts. Apart from some biological variability in the three tumors from each group, it appears that J54 (or CPT + J54) increased the pRAD54 signal rather than its expected decrease (Appendix A). While this may seem like a contradiction of our study, the fact is that the phosphorylation of pRAD54-T700 primarily alters its nucleoplasmic relocalization from the cytoplasm, where it predominantly resides in the absence of DNA damage (DSBs), and we observed that the nuclear fraction remains unaltered by the addition of J54, likely because of the absence of a nuclear phosphatase [32]. Indeed, our fractionation studies revealed that after IR, pRAD54-T700 almost exclusively relocalizes to the nuclei, based on fractionation experiments. We further demonstrated that even treatment with J54, an inhibitor of TLK1, reduces the total cell extract pRAD54 signal; the pT700-nuclear fraction remains unaltered by the addition of J54 [32]. In this specific context in mice, prolonged chronic treatment with J54 or (the more relevant) CPT + J54 may result in unexpected levels of pRAD54-T700. Since our working hypothesis is that CPT eventually results in the generation of DSBs that require processing via RAD54/HRR, but also that J54 alone can result in the accumulation of DNA damage in vivo due to the hypoxic tumor microenvironment, this could be an explanation for the rather unexpected findings of the increased pRAD54 signal in two of the J54(±)CPT tumor samples.

To ensure that J54 inhibited TLK1 within those tumors, despite the lack of evidence using pRAD54-T700 as a reader, we checked the same tissue extracts for pNEK1-T141, our most reliable relay for TLK1 activity. Our pNEK1-T141 Ab preferentially detects the phosphorylated form, which runs higher up on SDS/PAGE, but also, less efficiently, the “lower” un-phosphorylated form (previously demonstrated via the overexpression of a T141A mutant [39]). As shown in Appendix A, all representative tumors from all the mice that included treatment with J54 display a “band shift” change toward the un-phosphorylated NEK1 form. In contrast to the apoptotic hallmark in the tumor tissue from the treated mice, which alone could explain the obvious size regression in the CPT + J54 combination group, as well as in the other groups to a lesser extent, the proliferative capacity (measured based on the PCNA expression as an indicator of the % of dividing cells) was not affected in any of the three tumors from each group (Figure 4). We conclude that heightened apoptotic induction, rather than the inhibition of proliferation, is the key explanation for the tumor cytoreduction throughout treatment.

Combining CPT with J54 significantly decreases OS with lesser body weight loss. A detailed analysis of the tumor growth/regression studies was conducted for statistical significance as described in the Section 2. Selected illustrations of these analyses are shown in Figure 5 and Appendix A. To summarize these observations, we observed a reduction in tumor volumes only at concentrations of CPT alone that also resulted in toxicity, as indicated by a loss of body weight >15% that necessitated the termination of the experiment before the established end-point for the establishment of tumor reductions. While the combination of CPT (6 mg/kg) and J54 resulted in significant weight loss, this was not so for the lower CPT concentrations. In addition, we confirmed a significant statistical interaction for tumor volume change vs. control and in OS when CPT and J54 were combined. The current study demonstrated a statistically significant CPT effect on TV. Without J54, the higher the CPT, the smaller the TV (*p* = 0.0045). With J54, the TV had an apparent increase (*p* < 0.0382), which, however, was revealed at necropsy to be contributed to by the replacement of tumor mass with adipose tissue (see Section 4).

## 4. Discussion

The concept of synthetic lethality has evolved around the central tenet that, by affecting two pathways, each one of which is not essential for viability, the resulting combination is engendering a non-viable outcome, either via a genetic approach or through pharmacologic targeting. Perhaps its most successful use in cancer therapy is illustrated by the application of PARPis, sometimes combined with other drugs, for managing gynecological malignancies [37] or PCa [40,41,42], particularly in the context of a BRCAness cancer presentation.

In this work, we have somewhat expanded on the concept of synthetic lethality by taking advantage of the knowledge of the repair process of CPT-induced lesions (a common chemotherapy) known to yield DSBs that are repaired via HRR (a key pathway for their accurate repair) which would otherwise result in lethality from unrepaired damage or engender a combination of genomic rearrangements that, in most cases, would produce non-viable configurations [43]. Specifically, we exploited our knowledge of a regulatory pathway that we recently uncovered, governing the activity of the key HHR protein, RAD54, by the kinase TLK1 [32]. RAD54 performs critical multiple functions during the HRR process [33], and both its activity and cellular localization were found to be regulated differently by TLK1 during the sequential phases of the process. Therefore, our strategy was to treat two of the most common CRPC human cell lines with progressively more tolerable doses of CPT in combination with J54, a relatively new TLK1 inhibitor [34]. After a relatively rapid verification, in vitro, of the proof of principle of this strategy by determining the viability of these cells, which are normally quite resistant to CPT [44], but not when challenged in combination with J54, we then progressed to the more critical xenograft model. For this more critical analysis, we used two methods to follow tumor growth or regression over time: the direct measurement of tumor diameters with calipers and IVIS imaging at the midpoint (~1 month) and endpoint (day before sacrifice) of the time course. Direct visual evidence of tumor regression for the most representative group (3 mg/kg of CPT+ J54) at the mid-point is displayed in Appendix A, which is best appreciated when viewed in comparison with the endpoint image. Of particular importance is, again, the radiance scale that, at the treatment midpoint, was 1.89 × 10^8^ (min) to 3.64 × 10^9^ (max), showing that the tumors were up to 5 logs more emissive (larger number of cells) than at the endpoint shrinkage stage. However, even at the midpoint, the radiance was already ~1 log lower than that of the control group. The lack of a simple correlation between the phosphorylation of RAD54-T700 (Appendix A) and the tumor regression in the mice treated with CPT + J54 made it difficult to mechanistically conclude that the apoptotic response seen in vivo was all due to a lapse in HRR during the processing of ICLs. There could be other possible explanations for the effect observed, as the inhibition of TLK1 can suppress tumor growth via other mechanisms.

Of interest are the observations we made about the effect of J54 on the weight and overall health appearance of the mice throughout the experiment, with the exclusion of the two 6 mg/kg CPT groups, for which the toxicity was excessive after six doses. Starting from the obvious observation that the mice treated with 3 mg/kg of CPT progressively lost weight, whereas those that also received J54 did not and appeared healthier and more active, we noted some peculiar differences in the adipose tissue. The mice treated with 3 mg/kg of CPT alone showed either cytostatic or modest cytoreduction of the tumors, but without an obvious infiltration of adipose tissue; the mice that also received J54 displayed a noticeable replacement of the shrunk tumor space with an overgrowth of adipocytes. We suggest that this may be due to the combination of two factors: (1) There is already well-described research within the literature on the interplay between cancer cells and locally adjacent adipocytes, which secrete cross-stimulatory adipokines and cytokines [45]. (2) It is well known that several phenothiazine (PTH) antipsychotics (as well as newer atypical ones) cause the accumulation of white fat tissue and some weight gain as an unfortunate side effect [46,47]. J54 is structurally a PTH, although it was specifically designed to have minimal binding to the D2R, explaining its low antidopaminergic properties [34], and such activity is held to be the main mechanism responsible for adipogenesis stimulation. Regardless, for PTHs, other mechanisms have been proposed, such as the increased expression of *sterol regulatory element-binding protein* (*SREBP*) and *very low-density lipoprotein* (*VLDL*) genes (rev. in [47]), which could be alternative explanations for the effect of J54 on the localized adipogenesis and overall maintenance of body fat/weight in the face of mice treated with CPT alone. These mice progressively lost some weight and were visibly emaciated during treatment, likely from the combination of tumor burden and toxicity from CPT, for either of which, cachexia is a major issue during the terminal phases of cancer progression.

Overall, we would like to conclude that the use of J54 to achieve pharmacologic synthetic lethality with CPT (one of the most common chemotherapeutic agents) has shown promising results in one model of a PCa xenograft, whereby the synthetic targeting of HRR after the generation of DNA adducts and ISLs was shown to improve the efficacy of the therapy in tumor regression greatly. An unexpected benefit was better general health and weight maintenance in the most effective dose combination group. While we do not yet know if there are other, more subtle side effects for J54 that, in the long run, may make its potential clinical use unattainable, so far, and at least in mice, we never lost an animal, nor did we see altered behaviors in those treated with a dose of up to 20 mg/kg. Treatment with radiation (XRT) is also postulated to respond favorably in combination with J54.

## Figures and Tables

**Figure 1 biomedicines-11-02987-f001:**
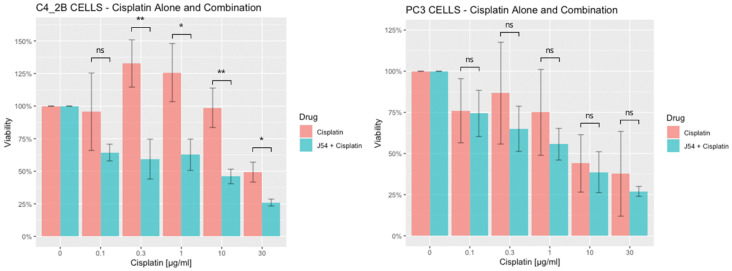
Viability of PC3 and C4-2B cells treated with CPT or in combination with J54. For each CPT dose, the combination with J54 was highly significant for C4-2B (mean *p* = 0.006, one-tailed paired) but marginal for PC3 cells. * *p* < 0.005; ** *p* < 0.001, ns = not significant.

**Figure 2 biomedicines-11-02987-f002:**
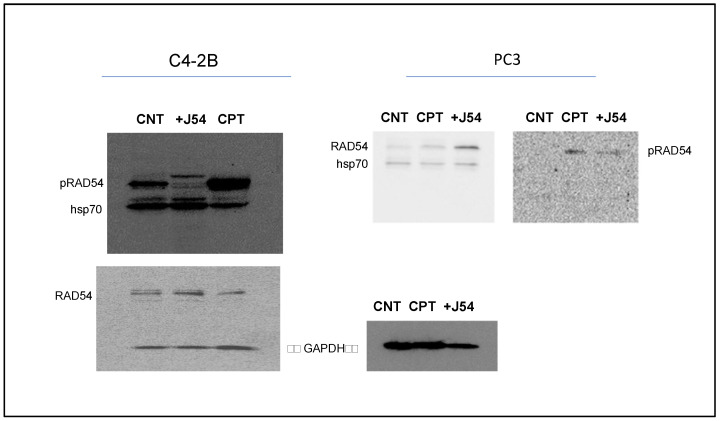
RAD54-T700 phosphorylation is induced by DNA damage (CIPT) w/o the induction of hsp70 expression, but is limited by concomitant J54 treatment.

**Figure 3 biomedicines-11-02987-f003:**
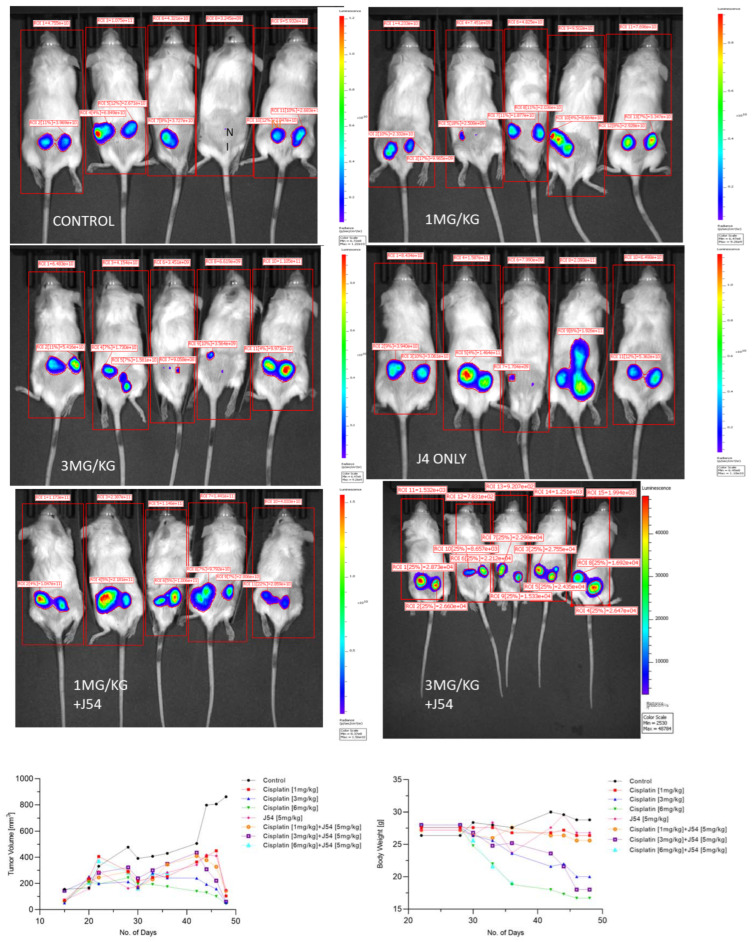
Time course of tumor progression in PC3-Luc xenografts. The pattern of tumor growth was studied over time with calipers (**bottom left**) or through one final analysis with the IVIS system one day before sacrifice. Body weights in time lapse were also recorded (**bottom right**). NI is not inoculated. Also note the important parameter of radiance divided by the full body area (label on top of rectangles) which, in essence, represents the total burden of tumor cells for the animal.

**Figure 4 biomedicines-11-02987-f004:**
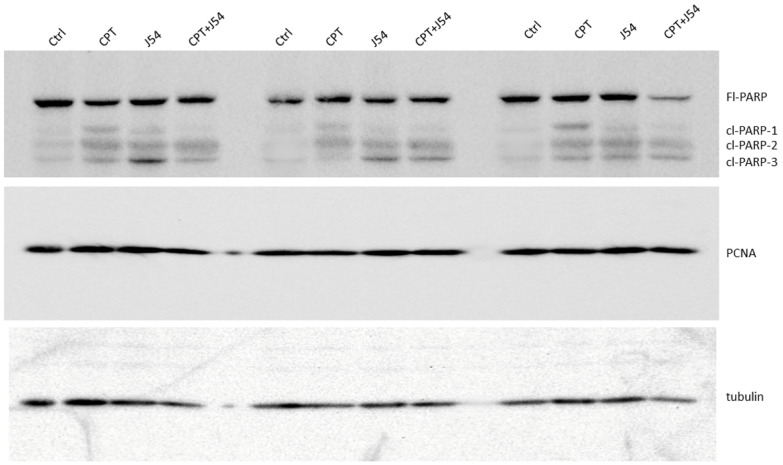
Western blot analysis of PARP and cleaved PARP from three sets of independent tumors from all treatment groups, and a separately probed one for PCNA and tubulin for loading control.

**Figure 5 biomedicines-11-02987-f005:**
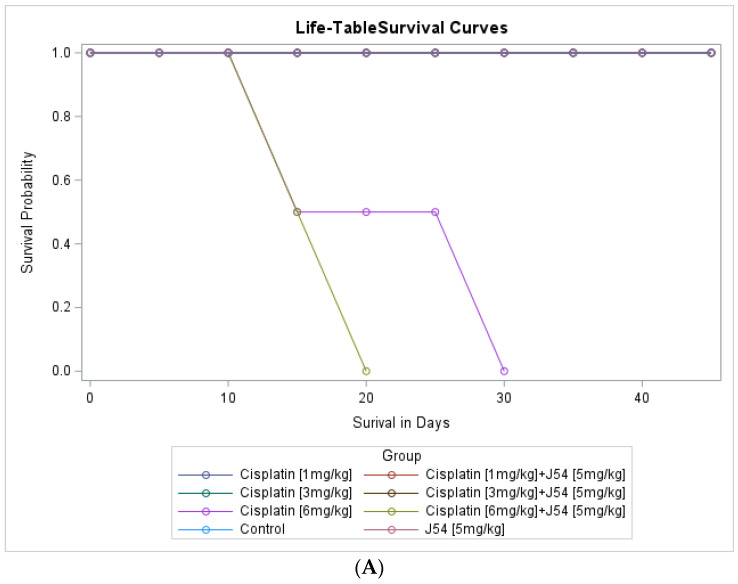
Key elements of statistical analyses for the combination CPT and J54 on tumor volumes and body weight changes vs. control. (**A**) Survival analysis. (**B**) Interaction analysis (a measure of synergy) between CPT and J54. (**C**) Interaction analysis for area under the curve (AUC) of tumor volumes over time and CPT doses.

## Data Availability

All data generated is included in the manuscript and available upon request.

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
