# Peer review of "Exploiting TLK1 and Cisplatin Synergy for Synthetic Lethality in Androgen-Insensitive Prostate Cancer"

_biomedicines, 2023, doi:10.3390/biomedicines11112987_

Round 1

Reviewer 1 Report

Comments and Suggestions for Authors

The article by Bhoir is interesting to see by Exploiting TLK1 to increase Cisplatin Synergy in Androgen-Insensitive Prostate Cancer. However, there are a few concerns that need the author's attention.

The cell lines used In this study, PC3 and C42B are androgen-independent. Any rationale for selecting these cell lines? It would be better to perform the combination efficacy using an AR cell line.

The combinational efficacy in both the cell lines looks pretty minimal. Is it significant between individual treatment versus the combination of cisplatin and J54?

The tumor growth inhibition in vivo correlates with body weight loss after 40 days. Any explanation for this effect just after 40 days.

Figure 4 Western blot did not add much information to the conclusion since the cleavage of PARP is not to that extent; it is recommended to repeat the experiment with another antibody for cl. PARP.

The tumor measurement and growth inhibition were shown for 50 days, whereas the survival curve was only for 30 days. Including the survival data for the remaining 20 days would be better.

Overall, the difference between in vitro and in vivo varies a lot. Any explanation?

Comments on the Quality of English Language

Please proof read for typos

Author Response

The cell lines used In this study, PC3 and C42B are androgen-independent. Any rationale for selecting these cell lines? It would be better to perform the combination efficacy using an AR cell line.

PC3 cells do not express the AR, but C42B overexpress the AR, which is probably why they can use autocrine-produced low levels of androgen for their growth.  The use of J54 for androgen-sensitive PCa lines was already studie in the iScience paper.

The combinational efficacy in both the cell lines looks pretty minimal. Is it significant between individual treatment versus the combination of cisplatin and J54?

Stats of significance were added to the figure 

The tumor growth inhibition in vivo correlates with body weight loss after 40 days. Any explanation for this effect just after 40 days.

The toxicity at lower doses of CPT did not result in rapid weight loss.  In the groups treated with 6mg/kg CPT the weight loss was rather immediate, which forced us to sacrifice those mice early on (after 3 weeks)

Figure 4 Western blot did not add much information to the conclusion since the cleavage of PARP is not to that extent; it is recommended to repeat the experiment with another antibody for cl. PARP.

We have included results for Cl-CA3 from an independent 2 sets of mice.

The tumor measurement and growth inhibition were shown for 50 days, whereas the survival curve was only for 30 days. Including the survival data for the remaining 20 days would be better.

The shorter survival period is only for the groups treated with 6 mg/kg CPT.  All the other groups survived till the established end-point of the experiment.

Overall, the difference between in vitro and in vivo varies a lot. Any explanation?

We should point out that in vitro, we analyzed viability 1 day later by the MTT assay. Preliminary clonogenic assays were also carried out, which demonstrated inhibition by CPT at lower concentrations than in the 1-day assay.  But we wanted to focus on early toxicity from persistent CPT (+/-J54) and not long-term survival after limited exposure to CPT. So, it’s hard to compare a high persistent dose in vitro treatment with a low-dose, chronic treatment in vivo.

Reviewer 2 Report

Comments and Suggestions for Authors

Authors: Siddhant Bhoir, Oluwatobi Ogundepo, Xiuping Yu, Runhua Shi, Arrigo De Benedetti 

Title:  Exploiting TLK1 and Cisplatin Synergy for Synthetic Lethality in Androgen-Insensitive Prostate Cancer 

COMMENTS: 

The Authors describe their interesting study on in vitro and in vivo models of prostate cancer in which they have achieved the enhanced beneficial effects by combining cisplatin with the TLK1 inhibitor, J54. The manuscript is well written and illustrated. I have only minor remarks: 

1. The section Materials and methods should contain detailed information about sources of both inhibitors, cisplatin and J54, used in this study. I mean producers and/or providers of either drug. In the manuscript, there are no refs regarding J54, why? Who does synthesize and provide this inhibitor? 

2. Taking into consideration that radiation exposure, like cisplatin, damages DNA in cancer cells, it would be nice to speculate about potential perspectives of J54 under combination of this inhibitor with radiotherapy of prostate cancer. 

Author Response

  1. The section Materials and methods should contain detailed information about sources of both inhibitors, cisplatin and J54, used in this study. I mean producers and/or providers of either drug. In the manuscript, there are no refs regarding J54, why? Who does synthesize and provide this inhibitor? 

CPT was from Millipore (Sigma) and this was added to the Methods.  J54 was synthesized by our group as described in the STAR-methods of the iScience paper.  However, we are aware that it is sold now by Probechem TLK1 inhibitor J54 | TLK1 inhibitor | Probechem Biochemicals

  1. Taking into consideration that radiation exposure, like cisplatin, damages DNA in cancer cells, it would be nice to speculate about potential perspectives of J54 under combination of this inhibitor with radiotherapy of prostate cancer. 

Added.  This was actually partly addressed in our earlier G&C paper when using less specific PTHs.

Round 2

Reviewer 1 Report

Comments and Suggestions for Authors

Sincere thanks to the authors for the quick review. However, some of the earlier reviews still need to be answered.

Considering the effect of J54+CIS combination in PC3 and C42B cells, in vitro data support better combination efficacy in C42B cells. However, for in vivo, the author's team selected PC3 cells. Any rationale?

Is the effect different in AR-positive versus negative cell lines? 

Figure 4 Western blot did not add much information to the conclusion since the cleavage of PARP is not to that extent; it is recommended to repeat the experiment with another antibody for cl. PARP. The comment stays - as the PARP cleavage product mostly give rise to 89 and 25 KDA product. The current blot with multiple bands brings back the concern that the observed product might be a non-specific product rather than the PAPR product. 

The tumor measurement and growth inhibition were shown for 50 days, whereas the survival curve was only for 30 days. Including the survival data for the remaining 20 days would be better. The authors can expand the survival graph until the experimental endpoint to capture the overall picture.  

Author Response

We appreciate the comments and should apologize for not having explained things a little better in the previous resubmission:

  1. We have added in the Fig. SI2 a new WB for cleaved Caspase 3, which gives essentially the same information as cleaved PARP in terms of another evidence of ongoing of apoptosis.
  2. In vivo experiments of tumor regression are lengthy, labor-intensive and quite expensive. Virtually nobody uses more than one model, particularly for IVIS-coupled studies. We happened to have PC3-Luc cells, and not C42B-Luc.  These can be made of course, but it will take an additional 1-2 months, apart from the new xenograft studies.  Also, as I tried to explain in the response, there is no good way to predict outcome in xenografts using (tolerable dose) CPT in chronic treatment based on data from high-dose/short duration experiments in vitro.  There was really no good way to know which model would fare better in vivo: AR+ or AR-.  But since there were careful studies of CPT treatment of PC3 xenografts (see link below) that showed high resistance, we opted for the more intractable model:

https://altogenlabs.com/xenograft-models/prostate-cancer-xenograft/pc-3-xenograft-model/

Another important point: PC3 display some feature shared by NEPC cells, including expression of chromogranin A.  Whereas for AR+ cancers there are some options, NEPC patients really have CPT as one of their remaining treatments available.  So, PC3 (AR- cells) were really a more valuable model for the xenograft studies.

  1. Possibly…but one could safely assume that CPT sensitivity is more complex and would depend on the strict characteristics of each cell line and that J54 will probably not alter that:

https://www.spandidos-publications.com/10.3892/ijo.2013.2223

  1. As stated in the response, all mice except from the 2 groups of 6 mg/kg CPT survived till the pre-established end-point. However, we have now prepared an alternative figure in accordance to the suggestion.

Round 3

Reviewer 1 Report

Comments and Suggestions for Authors

Authors team addresses the comments except for the multiple PARP cleavage products.